# Parallel Online Similarity Join over Trajectory Streams

## Abstract

Trajectory Similarity Join (TS-Join), as a fundamental operation in trajectory data analytics, has been extensively investigated by existing studies in data science community. However, existing solutions are almost designed for offline static trajectories, which cannot guarantee real-time feedback. In addition, the join results retrieved from existing solutions generally contains a large proportion of out-of-date similar pairs, making them inapplicable to evolving trajectories. In this light, we study a novel problem of online time-aware trajectory similarity join: Given a stream of evolving trajectories, we aim to dynamically discover trajectory pairs whose spatio-temporal similarity is no less than a specified threshold in a real-time manner. We innovatively introduce a time-aware exponential-decaying similarity function to eliminate out-of-date results. To support real-time querying over large populations of trajectories, we develop a Parallel Online Trajectory Similarity Join (POTSJ) framework incorporating with well-designed workload balancing techniques. We further enhance join efficiency through effective pruning strategies and tailored approximation techniques. The POTSJ framework we propose, which incorporates these elements, is capable of processing online TS-Join while simultaneously satisfying three key objectives: real-time result updates, comprehensive trajectory evaluation, and scalability. Extensive experiments on real-world datasets validate the efficiency and scalability superiority of our POTSJ framework in processing online TS-Join.

## 1 Introduction

Due to the increasing popularity of GPS-enabled devices and location-based services, the volume of trajectory data has experienced skyrocketing growth. Efficiently processing and analyzing large-scale trajectory data via fundamental operations such as trajectory similarity search [8, 12, 18, 24, 28] and similarity join [2, 3, 5, 7, 16, 17, 20, 26] have become a major research direction in the data science community. These efforts lay the foundations for a variety of location-based services, such as anomaly detection [23, 29], ride-sharing recommendation [18, 19] and location-based data cleaning [2].

In recent years, the maturation and widespread deployment of real-time data processing frameworks have led to a significant shift in the utilization of trajectories. The prevailing paradigm now emphasizes real-time collection, processing, and feedback. Researchers no longer focus solely on achieving effective offline batch processing of trajectories; instead, they increasingly leverage parallel computing resources to facilitate online computations with low latency and

high throughput as primary objectives. Some studies have developed in-memory online search algorithms for trajectory similarity queries and similarity joins [9–11, 14].

In existing trajectory similarity joins, the similarity between two trajectories is determined and not updated when their locations are not updated. All trajectory pairs whose similarities once exceed the given threshold are stored in the join result. As the processing clock (i.e., current timestamp) advances, the join results retrieved from these solutions inevitably contains a large proportion of out-of-date similar pairs, making them ineffective to evolving trajectories. To eliminate outdated results while maintaining real-time joins, online trajectory similarity joins are proposed.

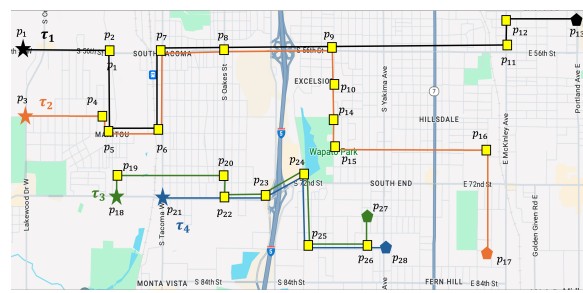

$\tau_1 = <p_1, 100>, <p_2, 105>, <p_5, 110>, <p_6, 115>, <p_7, 120>, <p_8, 125>, <p_9, 130>, <p_{11}, 135>, <p_{12}, 140>, <p_{13}, 145>.$

$\tau_2 = <p_3, 100>, <p_4, 105>, <p_5, 110>, <p_6, 115>, <p_7, 120>, <p_8, 125>, <p_9, 130>, <p_{10}, 135>, <p_{14}, 140>, <p_{15}, 145>, <p_{16}, 150>, <p_{17}, 155>.$

$\tau_3 = <p_{18}, 0>, <p_{19}, 5>, <p_{20}, 10>, <p_{22}, 15>, <p_{23}, 20>, <p_{24}, 25>, <p_{25}, 30>, <p_{26}, 35>, <p_{27}, 40>.$

$\tau_4 = <p_{21}, 10>, <p_{22}, 15>, <p_{23}, 20>, <p_{24}, 25>, <p_{25}, 30>, <p_{26}, 35>, <p_{28}, 40>.$

**Figure 1: Example of online trajectory similarity join**

Let us consider Figure 1 as an example. There are 4 trajectories with source locations marked with stars and destination locations marked with pentagons, denoted by $\tau_1$, $\tau_2$, $\tau_3$, and $\tau_4$. Assume that location samples of all trajectories are updated per 5 minutes. A time-stamped location sample consists of a location $p$ and a timestamp $t$ when this location is reported, denoted as a tuple $<p, t>$. We can observe that trajectories $\tau_3$ and $\tau_4$ are spatially and temporally similar to each other. Assume that similarity of $\tau_3$ and $\tau_4$ exceeds the given threshold when the processing clock advances to 40. As a result, the trajectory pair ($\tau_3$, $\tau_4$) is recorded in join result. Existing solutions do not update the similarity between $\tau_3$ and $\tau_4$ after clock advanced to 40, because no new location sample is further reported for $\tau_3$ and $\tau_4$. As a result, the trajectory pair ($\tau_3$, $\tau_4$) will be constantly included in the join result. However, it is meaningless and memory-consuming to maintain such pairs if their locations are not updated for a sufficiently long time. In contrast, their similarity is expected to decay as the processing clock progresses, and be excluded from the join result eventually. Let us consider $\tau_1$ and $\tau_2$ then. Before the processing clock advances to 130, they are almost spatially and temporally coincident. Assume that the similarity of $\tau_1$ and $\tau_2$ exceeds the given threshold when clock advances to 130. Even though $\tau_1$ and $\tau_2$ toward opposite direction after this moment, existing solutions may maintain this pair in the join result for a considerable long period. In online context, we

concern more with the similarity between the most recently updated trajectories. It is expected to eliminate pair $(\tau_1, \tau_2)$ from the result set after a short period after 130 due to their dramatic dissimilarity.

In this light, we define and study a novel problem of online Trajectory Similarity Join (online TS-Join): Given a stream of evolving trajectories, we target to dynamically discover all trajectory pairs whose spatio-temporal similarity is no less than a pre-defined threshold in a real-time manner. We incorporate exponential decaying into similarity function to meet the real-time requirements in online settings, thus addressing the deficiency confronted in the example above. We formally define a purposeful similarity metric to evaluate the similarity between two trajectories by taking into consideration all trajectory points both spatially and temporally. To serve applications well, online TS-Join must be efficient and must scale to massive populations. An effective online TS-Join approach must meet the following three requirements. (1) *Real-time updates.* As the trajectories evolve, the approach is able to dynamically update join results in a real-time manner. (2) *Comprehensive evaluation.* To ensure promising similarity joins, it is crucial to consider both the spatial and temporal proximity of all trajectory points. (3) *Scalability.* The approach scales to massive-scale evolving trajectories and ensure real-time result updates.

Existing studies on online trajectory search (e.g., [11, 14]) employ a sliding window to delimit a pre-defined time range over the trajectory stream and perform similarity joins within this window, thereby ignoring trajectory points outside this window. The query efficiency heavily relies on the window size. However, it is non-trivial to provide an exact window length beforehand in real-world applications. In cases like detecting co-movement patterns of animal migration or tropical cyclones, where trajectories are observed over a long period of time. Due to their limited scalability, they are computational prohibited if the window size is too large. As a result, they cannot support real-time updates or scale to massive-scale populations. Additionally, they do not consider the time-aware decaying of these outdated trajectory pairs. Therefore, their solutions cannot be directly used to solve our problem under exponential-decaying setting.

To answer the online TS-Join while ensuring real-time updates, comprehensive evaluation, and scalability simultaneously, we propose a Parallel Online Trajectory Similarity Join (POTSJ) framework. First, we develop a matrix-based approach that allows for efficient parallel joins. Trajectory points are divided into several independent partitions. A two-stage load balancing mechanism is employed that ensures both dynamic load balance and the effectiveness of join results. Next, multi-level prune techniques are developed for incremental similarity computation, allowing both global and local prune when dealing with massive evolving trajectory data. Finally, we propose an approximate algorithm to optimize the computation process when merging trajectory similarities. The main contributions of this paper can be summarized as follows.

- We define and study a novel problem of online TS-Join, in which a time-aware exponential-decaying similarity function is tailored.
- We propose a Parallel Online Trajectory Similarity Join (POTSJ) framework to process online TS-Join while simultaneously ensuring real-time updates, comprehensive evaluation, and scalability.

- We devise a matrix-based partition scheme and propose a two-stage load balancing algorithm that ensure both dynamic load balance and the completeness of join results.
- We propose an incremental similarity calculation algorithm with multi-level pruning techniques.
- We develop an approximate algorithm to further improve join efficiency with high recall.
- Extensive experiments are conducted on two real-world datasets to demonstrate the efficacy and efficiency of the POTSJ framework.

## 2 Problem Statement

In this section, we cover some essential concepts, online trajectory similarity measurement and formal problem statement.

### 2.1 Preliminaries

**Definition 1: Time-stamped location.** A time-stamped location $p$ is defined as a tuple $(s, t)$, where $s$ is the spatial coordinate (e.g., longitude, latitude), and $t$ is the time when this location is visited.

**Definition 2: Trajectory.** A trajectory $\tau$ of a moving object $o$ is defined as a finite, time-ordered location sequence that can be represented as $\langle p_{(\tau,1)}, p_{(\tau,2)}, \ldots, p_{(\tau,|\tau|)} \rangle$, where $p_{(\tau,j)}$, $j \in \{1, 2, \ldots, |\tau|\}$ is the $j$-$th$ time-stamped location of trajectory $\tau$, and $|\tau|$ is the length of trajectory $\tau$.

**Definition 3: Trajectory location stream.** Given a trajectory set $T$, the trajectory location stream of $T$, denoted by $S_T$, is an unbounded, infinite sequence of sample locations, which consists of sample locations collected from various moving objects in an online fashion.

A trajectory location stream generated by moving objects $o_1$, $o_2$ and $o_3$ is $\langle p_{(\tau_1,1)}, p_{(\tau_2,1)}, p_{(\tau_3,1)}, \cdots, p_{(\tau_2,12)}, p_{(\tau_1,11)}, p_{(\tau_3,10)}, \cdots \rangle$. Note that sample locations collected from a specific moving object are time-ordered, while the order of sample locations in a stream collected from different moving objects may not be strictly preserved. For ease of presentation, we use $p_{\tau_i}$ to denote a sample location in trajectory $\tau_i$ when the context is clear.

### 2.2 Similarity Measurement

To focus on the similarity between the most recently updated trajectories, we incorporate a time-aware exponential-decaying multiplier into the trajectory similarity measure. The time-aware exponential-decaying multiplier is defined in Definition 4.

**Definition 4: Time-aware exponential-decaying multiplier.** Given a time offset $\Delta$ and an exponential-decaying factor $\lambda$, the time-aware exponential-decaying multiplier $f(\Delta)$ is defined by Equation 1.

$$f(\Delta) = e^{-\lambda \cdot \Delta} \quad (1)$$

Intuitively, a greater $\lambda$ and a greater $\Delta$ both result in faster decay.

**Definition 5: Location-to-Trajectory similarity.** Given a location $p_{\tau_i}$ and a trajectory $\tau_j$, the spatial distance $d_s(p_{\tau_i}, \tau_j)$ and temporal distance $d_t(p_{\tau_i}, \tau_j)$ between $p_{\tau_i}$ and $\tau_j$ are defined by Equations 2 and 3. On top of Equations 2 and 3, we define the spatial and temporal similarity between a location $p_{\tau_i}$ to a trajectory $\tau_j$, denoted as $Sim_s(p_{\tau_i}, \tau_j)$ and $Sim_t(p_{\tau_i}, \tau_j)$, which are defined by Equation 4 and 5, respectively.

$$d_s(p_{\tau_i}, \tau_j) = min_{p_{\tau_j} \in \tau_j} \left\{ ||p_{\tau_i}.s - p_{\tau_j}.s|| \right\} \quad (2)$$

$$d_t(p_{\tau_i}, \tau_j) = \min_{p_{\tau_j} \in \tau_j} \left\{ \left| p_{\tau_i}.t - p_{\tau_j}.t \right| \right\} \tag{3}$$

$$Sim_s(p_{\tau_i}, \tau_j) = f(\Delta_s(p_{\tau_i}, \tau_j)) \cdot e^{-d_s(p_{\tau_i}, \tau_j)} \tag{4}$$

$$Sim_t(p_{\tau_i}, \tau_j) = f(\Delta_t(p_{\tau_i}, \tau_j)) \cdot e^{-d_t(p_{\tau_i}, \tau_j)} \tag{5}$$

Here, function $e^{-d}$ maps distance to similarity while maintaining a negative correlation between them, constraining similarity to the range $(0, 1]$. Factors $\Delta_s(p_{\tau_i}, \tau_j)$ and $\Delta_s(p_{\tau_i}, \tau_j)$ are employed to measure the temporal offset of the latest $d_s(p_{\tau_i}, \tau_j)$ and $d_t(p_{\tau_i}, \tau_j)$ with respect to the current system clock, which is defined as follows:

$$\Delta_s(p_{\tau_i}, \tau_j) = t_{clock} - t_{update}(d_s(p_{\tau_i}, \tau_j)) \tag{6}$$

$$\Delta_t(p_{\tau_i}, \tau_j) = t_{clock} - t_{update}(d_t(p_{\tau_i}, \tau_j)) \tag{7}$$

where $t_{clock}$ is current system clock, and $t_{update}(d_s(p_{\tau_i}, \tau_j))$ and $t_{update}(d_t(p_{\tau_i}, \tau_j))$ are the latest update timestamps of $d_s(p_{\tau_i}, \tau_j)$ and $d_t(p_{\tau_i}, \tau_j)$ respectively. Under this definition, as the streaming computation progresses, those not updated location-to-trajectory similarity for a long term share a relatively small weight on the trajectory similarity calculations, and vice versa.

**Definition 6: Exponential-decaying trajectory similarity.** Given trajectories $\tau_i$ and $\tau_j$, the spatial and temporal similarities between them, denoted as $Sim_s(\tau_i, \tau_j)$ and $Sim_t(\tau_i, \tau_j)$, are defined as follows.

$$Sim_s(\tau_i, \tau_j) = \frac{\sum_{p_{\tau_i} \in \tau_i} Sim_s(p_{\tau_i}, \tau_j)}{|\tau_i|} + \frac{\sum_{p_{\tau_j} \in \tau_j} Sim_s(p_{\tau_j}, \tau_i)}{|\tau_j|} \tag{8}$$

$$Sim_t(\tau_i, \tau_j) = \frac{\sum_{p_{\tau_i} \in \tau_i} Sim_t(p_{\tau_i}, \tau_j)}{|\tau_i|} + \frac{\sum_{p_{\tau_j} \in \tau_j} Sim_t(p_{\tau_j}, \tau_i)}{|\tau_j|} \tag{9}$$

Note that spatial and temporal similarities are in the range $(0, 2]$. Finally, we linearly combine them to comprehensively measure the spatio-temporal similarity between two trajectories, denoted as $Sim_{st}(\tau_i, \tau_j)$, which is calculated by Equation 10.

$$Sim_{st}(\tau_i, \tau_j) = \alpha \cdot Sim_s(\tau_i, \tau_j) + (1 - \alpha) \cdot Sim_t(\tau_i, \tau_j) \tag{10}$$

Here, parameter $\alpha \in [0, 1]$ serves as a user-defined parameter adjusting the weights of spatial and temporal similarity. Note also that the range of $Sim_{st}$ is $(0, 2]$. The value of $Sim_{st}$ is symmetric (i.e., $Sim_{st}(\tau_i, \tau_j) = Sim_{st}(\tau_j, \tau_i)$).

Our modeling of trajectory similarity aligns with existing studies (e.g, [16]), which distinguishes itself by additionally imposing a time-aware exponential-decaying multiplier. Introducing exponential decaying allows us to focus more on those latest updated and generated trajectories when conducting online trajectory similarity join. Parameter $\lambda$ can be specified to application scenarios. When $\lambda$ is zero, the similarity measure degenerates to those used by offline settings with static trajectories. Note that our time-aware exponential-decaying multiplier can be extended to other popular similarity measurements in an online setting (e.g., DTW [25], EDR[4]).

## 2.3 Problem Statement

In this section, we formally define the problem of Online Trajectory Similarity Join (Online TS-Join). Let $P$ and $Q$ be two distinct trajectory sets.

**Definition 7: Online TS-Join.** Given two evolving trajectory location streams $S_P$ and $S_Q$, and a similarity threshold $\theta$, the online TS-Join dynamically maintain a set $A$ of all trajectory pairs from sets $P$ and $Q$ whose exponential-decaying trajectory similarity no smaller than $\theta$, i.e., $\forall(\tau_i, \tau_j) \in (P \times Q) \setminus A(Sim_{st}(\tau_i, \tau_j) \leq \theta)$. Note that the set $A$ is dynamically updated as trajectory locations are evolving.

Unlike offline trajectory join problems, the online trajectory similarity join problem defined in this paper is based on dynamically generated trajectory streams, in which trajectory location involves in a real-time, continuous and iterative fashion.

## 3 Framework Overview

Figure 2 illustrates an overview of our Parallel Online Trajectory Similarity Join (POTSJ) framework, which consists of three main components, namely workload partition & dynamic adjustment, incremental similarity update, and result merge.

For workload partition and balance, we use a matrix-based partition scheme to partition evolving trajectory location stream $S_P$ and $S_Q$. Each matrix element corresponds to a parallel thread conducting downstream similarity update. As shown in the first part, suppose the matrix size is $3 \times 3$, 9 threads are employed to update location-to-trajectory similarity incrementally downstream.

To enable incremental similarity updates, we introduce a module for dynamic workload adjustment, allowing real-time monitoring and redistributing new data to lower-load threads when imbalances occur. This ensures both data independence across partitions and completeness of the similarity join results. As the input stream evolves, location-to-trajectory similarities are incrementally updated in parallel across partitions. When a new sample location arrives, many location-to-trajectory updates are triggered within its partition. To eliminate unqualified trajectory pairs at an early stage, we implement a multi-level pruning technique that maintains upper bounds of spatio-temporal similarities. Trajectory pairs with upper bounds below the threshold $\theta$ are pruned globally. For the remaining pairs, local pruning strategies, including grid neighborhood search and dimensional linear pruning, are developed to further reduce unnecessary updates.

Finally, we merge the upstream updated location-to-trajectory similarity into spatio-temporal similarity (cf. Equations 8 and 9). As each processing clock update incurs exponential decay, which is computationally expensive. We propose an approximate algorithm to merge similarity by interval update. Note that similarity merge among trajectory pairs are independent and thus can be parallelized. Finally, by comparing $Sim_{st}$ with $\theta$, the final join results can be obtained.

We proceed to introduce the details of workload partition & balance, incremental similarity update, and result merge in Sections 4, 5, and 6, respectively.

## 4 Workload Partition & Dynamic Adjustment

An effective partition scheme is expected to achieve the following three fundamental objectives: (1) Independence, whereby the data within each partition should be sufficient to fulfill the computational task without inter-partition communication; (2) Completeness, ensuring that the merge of computation results from various partitions guarantees an accurate final outcome; (3) Balance, signifying that the data volume processed by each partition should be maintained as evenly as possible. A matrix-based partition scheme with a tailored dynamic workload adjustment technique is proposed to achieve the aforementioned objectives.

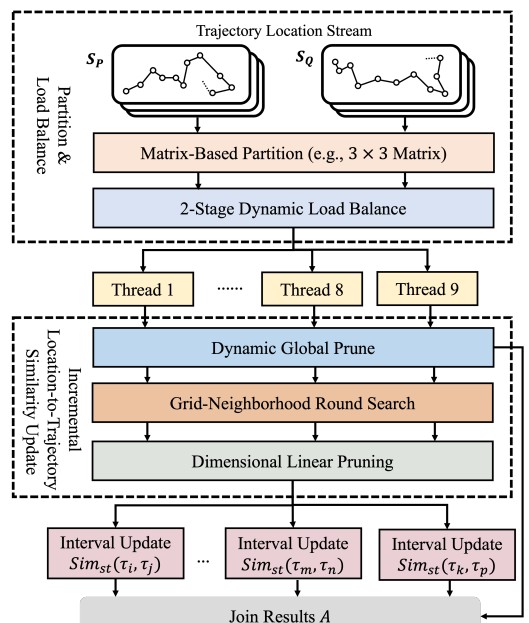

**Figure 2: Parallel online trajectory similarity join (POTSJ) framework**

## 4.1 A Matrix-Based Partition Approach

An existing solution [14] directly assigns sample locations from $S_P$ and $S_Q$ in a cyclic manner by column and row, achieving a natural balance regarding the number of trajectory locations in each partition. However, it may cause scattered sample locations in each partition and cannot satisfy independence and completeness requirements.

To address this, we propose a matrix-based partition scheme. Let #$Threads$ be the number of available threads for trajectory similarity join downstream. Assume that #$Threads$ is a perfect square. We construct a partition matrix with size $\sqrt{\#Threads} \times \sqrt{\#Threads}$, in which each element represents a partition. Each partition consists of two groups of location streams from a subset of $S_P$ and a subset of $S_P$, respectively. Unlike the partition approach in [14], we adopt trajectory as the fundamental unit for distribution, ensuring each partition contains entire trajectory for further validation. In our settings, locations in $S_P$ are distributed in a column-wise cyclic manner, while locations in $S_Q$ is distributed in a row-wise cyclic manner. For a trajectory $\tau \in P$, if any location of $\tau$ has already been distributed to the partitions in a certain column of the matrix before, then all subsequent locations of $\tau$ will be assigned to the partitions in the same column. Otherwise, we assign the locations of $\tau$ to the column based on two cases: if current partition matrix still results in balanced workloads, then we assign incoming locations in a cyclic manner. Else, we assign incoming locations to the column with the minimal workloads. The same is true when handling locations in $S_Q$.

Figure 3 illustrates a partition example with matrix size $2 \times 2$. Let trajectory sets $P$ be $\{(\tau_1, \tau_5), (\tau_3, \tau_7)\}$ and $Q$ be $\{(\tau_2, \tau_6), (\tau_4, \tau_8)\}$, which have been divided into 2 subsets. Given trajectory location streams $S_P = \langle p_{(\tau_1,1)}, p_{(\tau_3,1)}, p_{(\tau_5,1)}, p_{(\tau_7,1)}, \dots \rangle$, and $S_Q = \langle p_{(\tau_2,1)}, p_{(\tau_4,1)}, p_{(\tau_6,1)}, p_{(\tau_8,1)}, \dots \rangle$, a partition example can be easily obtained by

joining each two subsets of $P$ and $Q$. In our partition scheme, each trajectory pair $< \tau_i, \tau_j > (\tau_i \in P, \tau_j \in Q)$ is processed within one and only one partition (e.g., $< \tau_5, \tau_6 >$ is constantly validated in the last partition in Figure 3), which ensures independence.

The union of join results from all partitions equals the complete Cartesian product of trajectories from $S_P$ and $S_Q$, which ensures completeness. The matrix-based partition approach we propose, which incorporates these advantages, allows for efficient parallel processing while guaranteeing the result correctness.

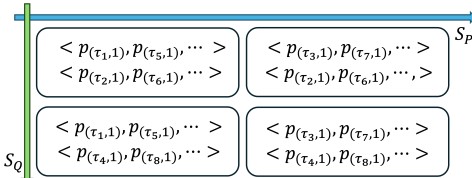

**Figure 3: Example of Matrix-based Stream Partition**

## 4.2 Two-Phase Dynamic Load Balance

While the number of trajectories within each partition is balanced, uneven load distribution between partitions arises when there exists a substantial disparity in the number of sample locations between trajectories due to different lifespans. However, the lifespan of a trajectory is unpredictable and cannot be determined in advance. To enhance workload balance, we introduce a two-stage adjustment algorithm, namely dynamic load monitoring and adjustment.

To consistently monitor the workload of each partition, we initialize a workload matrix $L$ with the same size as the partition matrix. Each individual matrix element records the load of corresponding partition. Once a new location is dispatched to a partition, the workload of this partition increases accordingly. To quantify the partition imbalance of columns and rows, we define two unbalanced factors $UBF_{col}$ and $UBF_{row}$ by Equation 11. Here, $i, j, t, k \in [1, \sqrt{\#Threads}]$. Notations $L_{col}[i]$ and $L_{row}[j]$ denote the sum of workloads within partitions in $i-th$ column and $j-th$ row of the load matrix, calculated as follows.

$$UBF_{col} = \frac{\max\{L_{col}[i]\}}{\min\{L_{col}[j]\}}; UBF_{row} = \frac{\max\{L_{row}[t]\}}{\min\{L_{row}[k]\}} \quad (11)$$

$$L_{col}[i] = \sum_{k=1}^{n} L(j, i); L_{row}[j] = \sum_{k=1}^{n} L(j, i), k \in [1, n] \quad (12)$$

Note that the unbalanced factors vary as the the influx of incoming trajectory locations. Once $UBF_{col}$ or $UBF_{row}$ reaches the given threshold $\epsilon$, the dynamic workload adjustments are triggered to ensure workload balance.

Specifically, if $UBF_{col}$ reaches the threshold $\epsilon$, we find the column with the minimum workload $L_{col}$. For a new trajectory $\tau_i \in P$, the subsequent incoming locations from $S_P$ will be assigned to all partitions in this column. We iteratively perform aforementioned assignment until $UBF_{col}$ falls below the threshold $\epsilon$. If $UBF_{row}$ reaches $\epsilon$, we find the row with the minimum workload $L_{row}$ to perform the aforementioned operations.

For an incoming location $p_{\tau_i}$ from $S_P$, there are three partition cases: (i) If $\tau_i$ has already been assigned to a certain column, then

$p_{\tau_i}$ is assigned to the partitions with the same column. Otherwise, (ii) If $UBF_{col}$ is less than $\epsilon$, $\tau_i$ and $p_{\tau_i}$ are distributed in columns in a cyclic manner. (iii). If $UBF_{col}$ is greater than $\epsilon$, both $\tau_i$ and $p_{\tau_i}$ are distributed to the column with the minimal workload.

The pseudo code of detailed dynamic workload adjustment algorithm is presented in Appendix C.1.

## 5 Incremental Similarity Update

Once an incoming location $p_{\tau_i} \in S_P$ is assigned to a certain column (or row), the trajectory similarity updates between $\tau_i$ and $\tau_j \in S_Q$ within this column (or row) are triggered. Specifically, two groups of similarity updates are required: **(a)** the spatial and temporal distance between $p_{\tau_i}$ and other trajectories $\tau_j$ within the partition, and **(b)** the spatial and temporal distance between sample locations $p_{\tau_j}$ belonging to other trajectories $\tau_j$ and the trajectory $\tau_i$ to which $p_{\tau_j}$ belongs. A straightforward approach involves traversing all sample locations within the partition, which is time-consuming when processing massive-scale locations.

To enable efficient similarity updates, we propose an incremental similarity update approach, in which some effective pruning strategies are employed. A grid-neighborhood round search and a dimensional linear pruning methods are proposed to process group updates **(a)** and **(b)**, respectively.

The pseudo code of overall incremental similarity update, spatial neighborhood search and dimensional linear prune is presented in Appendices C.2, C.3 and C.4 respectively.

### 5.1 Global Pruning Strategy

**Lemma 1.** Given any two trajectories $\tau_i$ and $\tau_j$, an upper bound of $Sim_s(\tau_i, \tau_j)$ is defined by Equation 13.

$$Sim_s(\tau_i, \tau_j).ub = \frac{\sum_{p_{\tau_i} \in \tau_i} e^{-d_s(p_{\tau_i}, \tau_j)}}{|\tau_i|} + e^{-min_{p_{\tau_i} \in \tau_i}\{d_s(p_{\tau_i}, \tau_j)\}} \quad (13)$$

Similarly, an upper bound of temporal similarity between two trajectories is defined by Equation 14. By combining $Sim_s(\tau_i, \tau_j).ub$ and $Sim_t(\tau_i, \tau_j).ub$, an upper bound of the spatio-temporal similarity between two trajectories is calculated by Equation 15. With this upper bound, we develop a global pruning strategy as follows: If $Sim_{st}(\tau_i, \tau_j).ub < \theta$, then pair $(\tau_i, \tau_j)$ cannot be a join result, thus we can safely prune $(\tau_i, \tau_j)$ at an early stage. Proof of Lemma 1 is presented in Appendix A.1.

$$Sim_t(\tau_i, \tau_j).ub = \frac{\sum_{p_{\tau_i} \in \tau_i} e^{-d_t(p_{\tau_i}, \tau_j)}}{|\tau_i|} + e^{-min_{p_{\tau_i} \in \tau_i}\{d_t(p_{\tau_i}, \tau_j)\}} \quad (14)$$

$$Sim_{st}(\tau_i, \tau_j).ub = \alpha \cdot Sim_s(\tau_i, \tau_j).ub + (1 - \alpha) \cdot Sim_t(\tau_i, \tau_j).ub \quad (15)$$

Let $\tau_i^*$ and $\tau_i$ denote the same trajectory before and after a new location $p_{\tau_i}$ is issued. In previous similarity computations, the distances between all locations in $\tau_i^*$ and all other trajectories within the same partition have already been calculated and cached, and can be directly retrieved. Assume that the number of trajectories within this partition is $N$. The time complexity of obtaining the upper bound $Sim_{st}(\tau_i, \tau_j).ub$ is $O(N)$, while calculating the exact similarity $Sim_{st}(\tau_i, \tau_j)$ requires $O(N^2)$ time. By leveraging our global pruning strategy, we can safely prune unqualified pairs at an early

stage, thus avoiding exact similarity computations and improving efficiency. Only for these few retained candidate trajectories, their exact similarities will be calculated downstream.

To obtain the the upper bound $Sim_{st}(\tau_i, \tau_j).ub$, the nearest location-to-trajectory distances $d_s(p_{\tau_i}, \tau_j)$ and $d_t(p_{\tau_i}, \tau_j)$ are required, in which we have to identify the nearest location of trajectory $\tau_j$ to the location $p_{\tau_i}$ within the same partition (cf. Equation 13). We proceed to introduce some fine-grained local prune methods to speed up this process.

### 5.2 Grid-Neighborhood Round Search

Recall that two groups of similarity updates are required if a new location $p_{\tau_i}$ is issued. In essence, the update process of group **(a)** is the nearest neighbor search from location $p_{\tau_i}$ to locations of other trajectories within the same partition. In this section, we propose a grid-neighborhood round search with multi-level pruning techniques to update group **(a)** efficiently.

Given a 2D spatial plane that represents underlying space of trajectory locations, we gradually split the plane into equal-size grid cells with a side length of $R$, which is based on the distribution of the input locations. Initially, we consider the first issued location, denoted as $p_c(x_c, y_c)$, as the central location. A central grid cell of dimension $R \times R$ is formed by extending from this central location in all four directions by $\frac{R}{2}$. Let $[[x_s, x_e], [y_s, y_e]]$ denote the boundaries of the central grid. As new locations are issued, iterative expansions along each of the four axes from the edges of this grid ultimately split entire spatial space into equal-size grids. Each distinct location falls exclusively within a unique grid. Specifically, for a new generated sample location $p(x_i, y_i)$, the boundaries of the grid to which $p$ belongs, denoted as $[[x_s', x_e'], [y_s', y_e']]$ is computed as follows.

$$x_s' = x_e + \left\lfloor \frac{x_i - x_e}{R} \right\rfloor \times R, x_e' = x_s' + R \quad (16)$$

$$y_s' = y_e + \left\lfloor \frac{y_i - y_e}{R} \right\rfloor \times R, y_e' = y_s' + R \quad (17)$$

Given a new issued location $p_{\tau_i}$, let $Set_T$ be the set of trajectory $\tau_j$ within the same partition as $p_{\tau_i}$. To update $d_s(p_{\tau_i}, \tau_j)$ efficiently, our two-phase approaches work as follows. Assume that $p_{\tau_i}$ falls within the grid cell $g$.

In Phase 1, a concentric round search based on grid units are conducted. Let $C$ be the number of concentric circles. During the period of round search, we begin with searching the grid $g$ containing $p_{\tau_i}$, then expand layer by layer until all grids within a square region centered at $g$ with side length $(2C + 1) \cdot R$ are evaluated. Through multi-level concentric round search, the minimum distances from $p_{\tau_i}$ to these evaluated trajectories are obtained. Assume that current search depth is $k$. We use $Set_k$ to denote the set of locations covered by the current search layer's grids. If there exists a location $p_{\tau_j} \in Set_k$ that satisfies $d_s(p_{\tau_j}, p_{\tau_i}) < d_s(g, p_{\tau_i}) + k \times R$, then $p_{\tau_j}$ is the location of $\tau_j$ closest to $p_{\tau_i}$. Here, $g$ represents the grid to which $p_{\tau_i}$ belongs, $d_s(g, p_{\tau_i})$ represents the minimum distance from $p_{\tau_i}$ to the four edges of $g$. $d_s(g, p_{\tau_i})$ is equivalent to the minimum radius of a circle centered at $p_{\tau_i}$ that is tangent to at least one edge of $g$. Note that once $\tau_j$'s closest location to $p_{\tau_i}$ has been identified, there is no need to search for the remaining locations of $\tau_j$, thus $\tau_j$ can be removed from $Set_T$. As a result, the number of trajectories that need to be further probed are continuously reduced during the process of round search.

In Phase 2, for these retained trajectories that are not probed in phase 1, one-to-one distance computation is required to seek the minimum distances from $p_{\tau_i}$ to these retained trajectories, which requires huge computational efforts, especially for long trajectories. To tackle it, we use a grid-based strategy to further prune candidates. Let $S_j$ be the set of grids that cover at least one locations of $\tau_j$. To compute the distance from $\tau_j$ to $p_{\tau_i}$, we calculate the radius of a circular region centered at $p_{\tau_i}$, ensuring it covers at least one complete grid in $S_j$. The closest location in $\tau_j$ to $p_{\tau_i}$ must fall within a certain grid $g \in S_j$ that overlaps with the circular region. We record these candidate grids, and search them one-by-one to find the nearest location $p_{\tau_j}$ to $p_{\tau_i}$.

Figure 4 demonstrates an example of our spatial grid-neighborhood search with the maximum depth $C = 3$. In Phase 1, we conduct a 3-round search centered on the blue grid where $p_{\tau_i}$ is located. The first round examines the blue grid, followed by searches of annular regions formed by green and orange grids. After each round, the nearest locations of these trajectories within red-circled grids are identified. In Phase 2, we focus on trajectories that have not been probed in Phase 1. For long trajectories, we record grids traversed by the trajectory as candidate grids (e.g., These grids fall within the black circle and shaded areas), where the nearest location may exist. Finally, we search for the nearest location to $p_{\tau_i}$ within these candidate grids.

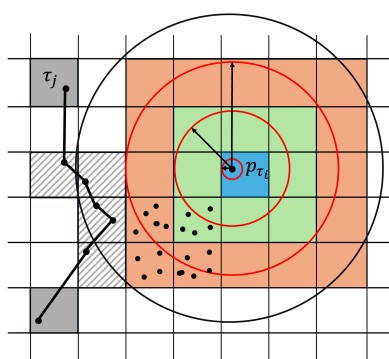

**Figure 4: Example of Spatial Grid-Neighborhood Search**

## 5.3 Dimensional Linear Pruning

In this section, we introduce a dimensional linear pruning strategy to update group (**b**) by effectively reusing previous computation results.

**Lemma 2.** Let $p_{\tau_i}^*$ be the closest location in $\tau_i$ to $p_{\tau_j}$ before the new location $p_{\tau_i}$ is issued. The midlocation $M$ of $p_{\tau_i}^*$ and $p_{\tau_i}$ lies on the perpendicular bisector $H$ of the segment $p_{\tau_i}^* p_{\tau_i}$, dividing the plane into two regions. For any $p$ on the side of $H$ near $p_{\tau_i}^*$, we have $||p, p_{\tau_i}^*|| < ||p, p_{\tau_i}||$, thus can be safely pruned.

Proof of Lemma 2 is presented in Appendix A.2. To implement this, we first perform a coarse-grained pruning at the grid level, removing all grids on the side of $H$ closer to $p_{\tau_i}^*$. Afterward, the remaining grid locations are verified for necessary updates.

Let us consider Figure 5 as an example of updating $d_s(p_{\tau_i}, \tau_i)$ when a new location $p_{\tau_i}$ arrives. There are two trajectories $\tau_i$ and $\tau_j$. We can observe that before $p_{\tau_i}$ is issued, $p_{\tau_i}^{*1}$ is the closest location of $\tau_i$ to $p_{(\tau_j,1)}$, $p_{(\tau_j,2)}$, and $p_{(\tau_j,3)}$, while $p_{\tau_i}^{*2}$ is the closest to the remaining

locations of $\tau_j$. Here, Perpendicular lines $H_1$ and $H_2$ from $p_{\tau_i}$ to $p_{\tau_i}^{*1}$ and $p_{\tau_i}^{*2}$ are calculated. According to the Lemma 2, only locations at the right of both $H_1$ and $H_2$ require updates (i.e., $d_s(p_{(\tau_j,7)}, \tau_i)$ and $d_s(p_{(\tau_j,8)}, \tau_i)$), while the shaded grids can be safely pruned.

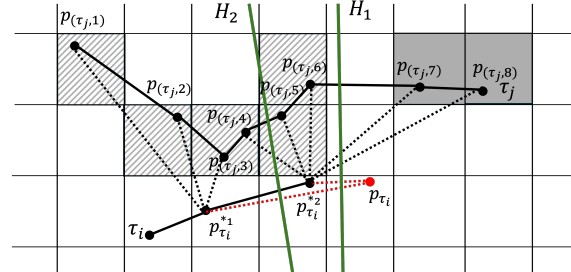

**Figure 5: Example of Spatial Linear Pruning**

Similarly, for a new issued location $p_{\tau_i}$, a one-dimensional linear pruning approach is applied to update temporal distance $d_t(p_{\tau_j}, \tau_i)$ based on Lemma 3.

**Lemma 3.** A lower bound and an upper bound for $p_{\tau_i}$'s timestamp whose $d_t(p_{\tau_j}, \tau_i)$ needs to be updated are defined by Equations 18 and Equation 19, respectively, where $p_{(\tau_i,k)}.t <= p_{\tau_i}.t$ and $p_{(\tau_i,k)}.t > p_{\tau_i}.t$. Here, $p_{(\tau_i,k)}$ represents the other sample locations in $\tau_i$, excluding the newly arrived location $p_{\tau_i}$.

$$(p_{\tau_j}.t).lb = \max_{p_{(\tau_i,k)} \in \tau_i} \left( \frac{p_{\tau_i}.t - p_{(\tau_i,k)}.t}{2} \right) \tag{18}$$

$$(p_{\tau_j}.t).ub = \min_{p_{(\tau_i,k)} \in \tau_i} \left( \frac{p_{(\tau_i,k)}.t - p_{\tau_i}.t}{2} \right) \tag{19}$$

Proof of Lemma 3 is presented in Appendix A.3. Based on Lemma 3, trajectory locations with timestamps less than the lower bound or exceeds the upper bound can be safely pruned.

## 6 Similarity Decaying & Merge

In Section 5, we propose several pruning methods to update Location-to-Trajectory similarities. Next, we need to compute decaying factor for each location-to-trajectory similarity pair and merge it to the spatio-temporal similarity defined as Equation 10. This helps determine if non-pruned trajectory pairs meet the similarity threshold.

### 6.1 Exact Algorithm for Similarity Merge

To obtain the trajectory similarity, we merge spatial and temporal Location-to-Trajectory similarities as defined in Equations 8 and 9.

With exponential decaying, each new tuple $< p_{\tau_i}, \tau_j, sim >$ from an upstream operator updates the similarity $Sim_{s/t}(p_{\tau_i}, \tau_j)$, affecting the similarity between trajectories $\tau_i$ and $\tau_j$. As new data arrives, the decaying factors for all Location-to-Trajectory pairs in the current partition also change due to the system clock's progression.

Thus, with the arrival of each new data, all Location-to-trajectory similarities must be traversed, decaying factors recalculated, and trajectory-to-trajectory similarities updated. This requires re-computation of all trajectory similarities from scratch, potentially causing a performance bottleneck.

## 6.2 Approximate Algorithm by Interval Update

Our approach divides the time axis into equal intervals, assigning data to the corresponding interval based on their entry time. Each interval uses its midlocation as the representative timestamp, replacing their individual timestamps. As time progresses, the decaying factors of older intervals approach zero. To handle data overflow, we set a small threshold, like $10^{-9}$, for interval expiration. When an interval's decaying factor drops below this threshold, it expires, and the data within it becomes invalid.

The decaying factor is used to reduce the influence of older data on current similarity calculations. By expiring intervals, we eliminate the impact of early records on trajectory similarity. The expiration threshold is adjustable; setting it to zero prevents expiration and ensures complete similarity results and comprehensive trajectory evaluation. With each timestamp advance, only valid intervals need updating, keeping the number of intervals small and fixed. In contrast, the exact algorithm requires updating $\#SimPairNum$, which grows over time. The pseudo code of similarity merge by interval update is presented in Appendix C.5.

## 7 Experiments

### 7.1 Experiment Settings

*Datasets.* We use two real-world datasets: the Beijing Taxi Dataset (BTD) [27] and the New York Taxi Dataset (NTD)[1]. BTD includes 10,357 trajectories with about 17 million sample locations collected over 7 days at a rate of 1,686 locations per minute. NTD contains 2 million trajectories and approximately 4 million locations over 45 days, with a sampling rate of 61 locations per minute. BTD mainly features long trajectories, while NTD has mostly short ones.

*Evaluation Metrics.* We use the following metrics to evaluate our model's efficiency and effectiveness: (a) Average Latency, the average time taken to compute and return a result for each trajectory pair after reading a location. (b) Average Throughput, the average number of trajectory pairs processed per unit of time. (c) Accuracy, the ratio of the spatio-temporal similarity $Sim_{st}$ calculated with interval updates to the exact $Sim_{st}$, measuring the accuracy loss due to interval updates.

*Compared methods.* We evaluate the performance of four methods: three iterative versions of our proposed framework and a variant of the Ghost framework from [11], the SOTA online trajectory similarity search framework:

- Ori-POTSJ: Straightforward method for online TS-Join.
- LP-POTSJ: POTSJ with only local prune techniques (Sections 5.2 and 5.3)
- GP-POTSJ: Full POTSJ framework with both global and local prune techniques (Sections 5.1, 5.2 and 5.3)
- Ghost*: An extended version of the original Ghost framework adapted for our problem. Ghost* removes the window constraint to ensure both real-time performance and completeness by considering all trajectory points. Additionally, we integrated a new module to support for the spatio-temporal similarity metric used here.

[1]https://data.cityofnewyork.us/Transportation/2016-Green-Taxi-Trip-Data/hvrh-b6nb/about_data

*Implementation.* All algorithms were implemented in Java using the Flink 1.13.0 stream processing framework for parallel execution. Experiments ran on a server with two Intel Xeon Gold 5128R processors (2.10 GHz) and 128 GB of memory, with 60 task slots sharing the memory. All trajectory data, index structures, and intermediate results were memory-resident. Results are averaged over multiple independent runs. Our model's code is available [2]. The parameter settings are listed in Table 1.

**Table 1: Parameter Settings**

|  | BTD | NTD |
|---|---|---|
| $|P|$ | 1M-4M/ default 1M | 0.5M-2M/ default 1M |
| $R$ | 0.001-0.007/default 0.001 | 0.001-0.007/default 0.001 |
| $\theta$ | 1.80-1.95/default 1.90 | 1.00-1.95/default 1.90 |
| $\alpha$ | 0.3-0.9/ default 0.5 | 0.3-0.9/ default 0.5 |
| $\lambda$ | 0.50-1.25/default 0.50 | 0.50-1.25/default 0.50 |
| $\sigma$ | 0-$10^{-4}$/ default $10^{-9}$ | 0-$10^{-4}$/ default $10^{-9}$ |
| Thread Count | 24-60/ default 60 | 24-60/ default 60 |

### 7.2 Efficiency Study

*Effect of cardinality of trajectory locations $|P|$.* As the number of trajectory locations $|P|$ increases, the computational cost for similarity join rises. To evaluate the model's efficiency, we vary the dataset size and measure the average latency and throughput of four baseline methods. Figures 6(a) and 6(c) show that for both NTD and BTD datasets, average latency increases with dataset size, but GP-POTSJ exhibits a significantly smaller increase and is over a hundred times more efficient than the other baselines. LP-POTSJ is also several times more efficient than Ori-POTSJ and Ghost*, with lower latency growth as $|P|$ increases. Ori-POTSJ and Ghost* perform similarly since they lack optimizations for similarity computation and suffer rapid efficiency degradation with larger datasets.

Figures 6(b) and 6(d) demonstrate that GP-POTSJ outperforms the other methods in throughput. However, due to the predominance of short trajectories in NTD, the global and grid-based local pruning methods used by LP-POTSJ and GP-POTSJ are less effective, reducing their efficiency compared to when processing BTD.

*Effect of thread counts.* We evaluate the framework's scalability by varying the number of threads used for computational tasks and measuring the average latency and throughput. As shown in Figure 7, increasing the thread count reduces latency and improves throughput across all baselines, demonstrating efficient scalability.

Note that parameter sensitivity study is presented in Appendix B

### 7.3 Effectiveness Study

*Effect of interval expiration threshold $\sigma$.* Applying interval update introduces accuracy loss. To assess its impact, we vary the interval expiration threshold $\sigma$ while keeping the interval length fixed at 1000 ms and measure the effect on ST-similarity accuracy (defined in Section 7.1). Ori-POTSJ does not use interval updates, so this experiment focuses on comparing the accuracy of GP-POTSJ and LP-POTSJ. As shown in Figure 8, increasing $\sigma$ causes a slight decrease in accuracy, but it remains above 95%, which is acceptable.

[2]https://anonymous.4open.science/r/POTSJ-2828

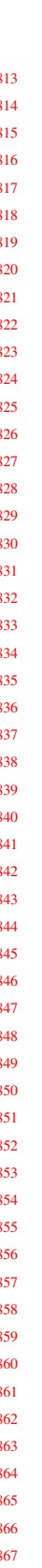

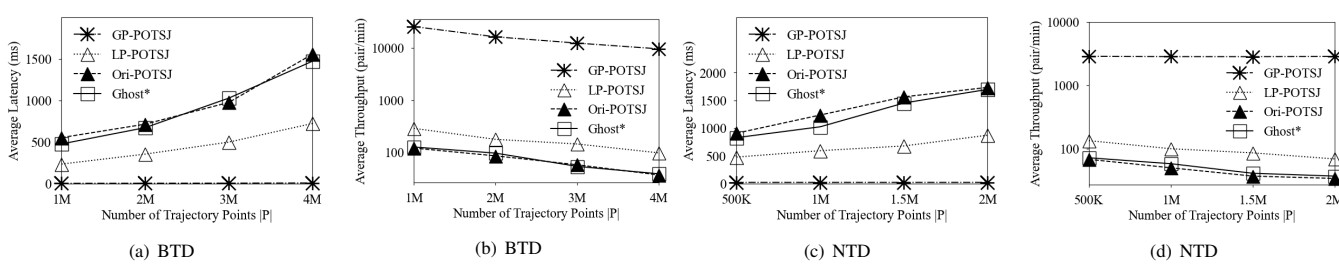

**Figure 6: Effect of cardinality of trajectory locations $|P|$**

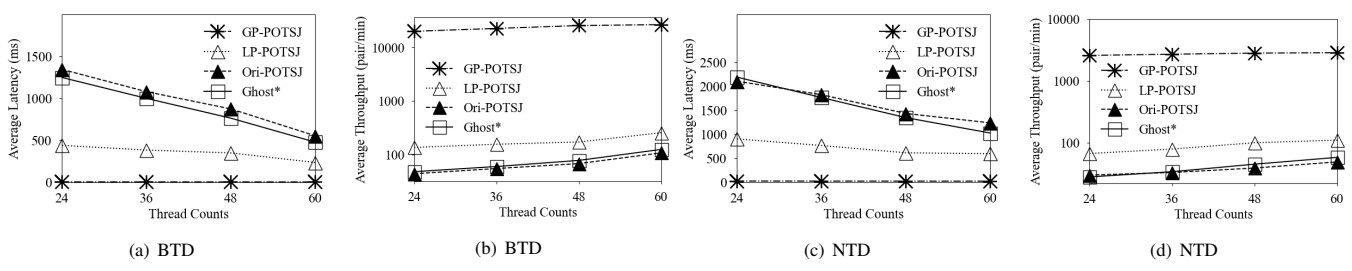

**Figure 7: Effect of thread counts**

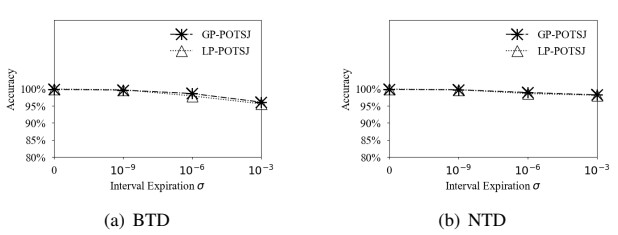

**Figure 8: Effect of interval expiration threshold $\sigma$**

## 8 Related Work

We review the literature regarding offline&online trajectory similarity analytics, and emphasize how our method distinguishes itself.

*Offline trajectory analytics.* Offline trajectory analytics [13, 20, 26], including similarity search [2, 6, 8, 12, 15, 18, 19, 24, 28] and similarity join [3, 5, 7, 16, 17, 21, 26] have been extensively studied. Shang et al.[16] proposed a two-phase parallel framework for spatial-temporal similarity join, employing deep pruning techniques for fast large-scale trajectory processing. In [20], they introduced a Spark-based in-memory framework with comprehensive APIs for trajectory similarity join. Yuan et al.[26] defined a trajectory similarity function based on road networks, using a filter-refine framework and a distributed memory system to achieve load balancing and efficient trajectory partitioning. Chen et al.[5] addressed semantic trajectory similarity join by incorporating textual data into a two-phase parallel matching approach. However, these methods rely on static, fixed trajectory datasets and are not suitable for dynamic, evolving trajectory scenarios, making them inapplicable to the problem addressed in this paper.

*Online trajectory analytics.* Online trajectory similarity analytics has gained popularity in recent years. Pan et al. [14] employed spatial distance measures such as Hausdorff [1], DTW [25], and LCSS [22], and proposed a matrix-based method to partition trajectory points evenly. Using window constraints, their algorithm identifies trajectories for similarity join but recomputes similarity from scratch each time the window slides, leading to redundant calculations. Fang et al. [11] support various spatial distance measures. Their framework also uses time windows but optimizes each measure and computes similarity incrementally, avoiding redundant recalculations. They propose a unified similarity definition and pruning techniques, significantly reducing time overhead. However, these window-based approaches are unsuitable for the problem addressed in this work because they compromise the completeness of join results and cannot provide fine-grained real-time feedback.

## 9 Conclusion

To the end, we propose the POTSJ framework to address a novel problem of online time-aware trajectory similarity join. By introducing a time-aware exponential decay factor into the spatio-temporal similarity function, we ensure real-time, dynamic updates that eliminate outdated results. Our matrix-based partitioning scheme, coupled with dynamic load balancing, effectively partitions trajectory streams while minimizing data redundancy. To enhance efficiency, we implement multi-level pruning techniques across both spatial and temporal dimensions, accelerating similarity calculations. Additionally, an approximate algorithm refines the merge of decaying similarity. Extensive experiments on real-world datasets validate that POTSJ achieves superior efficiency and scalability.

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

## A Theorem Proof

### A.1 Proof of Lemma 1

*Proof:* Let $p^*_{\tau_j}$ be the location in $\tau_j$ spatially closest to $p_{\tau_i}$. We have $d_s(p_{\tau_i}, \tau_j) = d_s(p_{\tau_i}, p^*_{\tau_j})$. According to Equation 2, we have $d_s(p^*_{\tau_j}, \tau_i) = \min_{p_{\tau_i} \in \tau_i}\{d_s(p^*_{\tau_j}, p_{\tau_i})\} \leq d_s(p_{\tau_i}, p^*_{\tau_j}) = d_s(p_{\tau_i}, \tau_j)$. It follows that $\forall p_{\tau_i} \in \tau_i, d_s(p_{\tau_i}, \tau_j) \geq d_s(p^*_{\tau_j}, \tau_i) \geq \min_{p_{\tau_j} \in \tau_j}\{d_s(p_{\tau_j}, \tau_i)\}$. By replacing $d_s(p_{\tau_i}, \tau_j)$ with $\min_{p_{\tau_j} \in \tau_j}\{d_s(p_{\tau_j}, \tau_i)\}$, we have:

$$Sim_s(\tau_i, \tau_j) \leq \frac{\sum_{p_{\tau_i} \in \tau_i} e^{-d_s(p_{\tau_i}, \tau_j)}}{|\tau_i|} + \frac{\sum_{p_{\tau_j} \in \tau_j} e^{-d_s(p_{\tau_j}, \tau_i)}}{|\tau_j|}$$
$$\leq \frac{\sum_{p_{\tau_i} \in \tau_i} e^{-d_s(p_{\tau_i}, \tau_j)}}{|\tau_i|} + e^{-\min_{p_{\tau_i} \in \tau_i}\{d_s(p_{\tau_i}, \tau_j)\}} \quad (20)$$

$\square$

### A.2 Proof of Lemma 2

*Proof:* Let the coordinates of location $p^*_{\tau_i}$ be $(x_1, y_1)$ and the coordinates of location $p_{\tau_i}$ be $(x_2, y_2)$. Therefore, the coordinates of the midlocation $M$ are $(\frac{x_1+x_2}{2}, \frac{y_1+y_2}{2})$. The equation of the perpendicular line $H$ can be established as: $y - \frac{y_1+y_2}{2} = -\frac{x_2-x_1}{y_2-y_1}(x - \frac{x_1+x_2}{2})$. Let location $A$ have coordinates $(x, y)$, and we can derive: $Ap_o^2 - Ap_i^2 = (x_1^2 - x_2^2) + (y_1^2 - y_2^2) + 2x(x_2 - x_1) + 2y(y_2 - y_1)$. For locations $A$ close to the side of $p_o$ or exactly on line $H$, if $y_1 \geq y_2$, the condition is satisfied: $y \geq -\frac{x_2-x_1}{y_2-y_1}(x - \frac{x_1+x_2}{2}) + \frac{y_1+y_2}{2}$. If $y_1 \leq y_2$, the condition is satisfied: $y \leq -\frac{x_2-x_1}{y_2-y_1}(x - \frac{x_1+x_2}{2}) + \frac{y_1+y_2}{2}$. Merging equations above, we obtain: $Ap_o^2 - Ap_i^2 \leq 0 \Rightarrow Ap_o \leq Ap_i$. $\square$

### A.3 Proof of Lemma 3

*Proof:* For $p_{\tau_j}$'s timestamps less than $LB$, there must exist $p_{(\tau_i,k)}$ where $p_{(\tau_i,k)}.t < p_{\tau_i}.t$, satisfying $p_{\tau_j}.t < \frac{p_{\tau_i}.t - p_{(\tau_i,k)}.t}{2}$. Therefore, $|p_{\tau_j}.t - p_{(\tau_i,k)}.t| < |p_{\tau_j}.t - p_{\tau_i}.t|$, and thus $p_{\tau_j}$ does not need to be updated and can be pruned. The proof for $UB$ is similar. $\square$

## B Parameter Sensitivity Study

*Effect of grid width R.* . We vary the grid width used for splitting the spatial plane to examine its impact on model performance. As shown in Figure 9, Ori-POTSJ, which lacks grid-based local pruning, is unaffected by the parameter $R$. However, LP-POTSJ and GP-POTSJ, which utilize local pruning, are significantly sensitive to grid width. On BTD (Figures 9(a) and 9(b)), as grid width increases, the average latency of LP-POTSJ and GP-POTSJ initially decreases but then sharply rises, while throughput first improves and then drops. A similar, though less pronounced, pattern occurs on NTD, due to

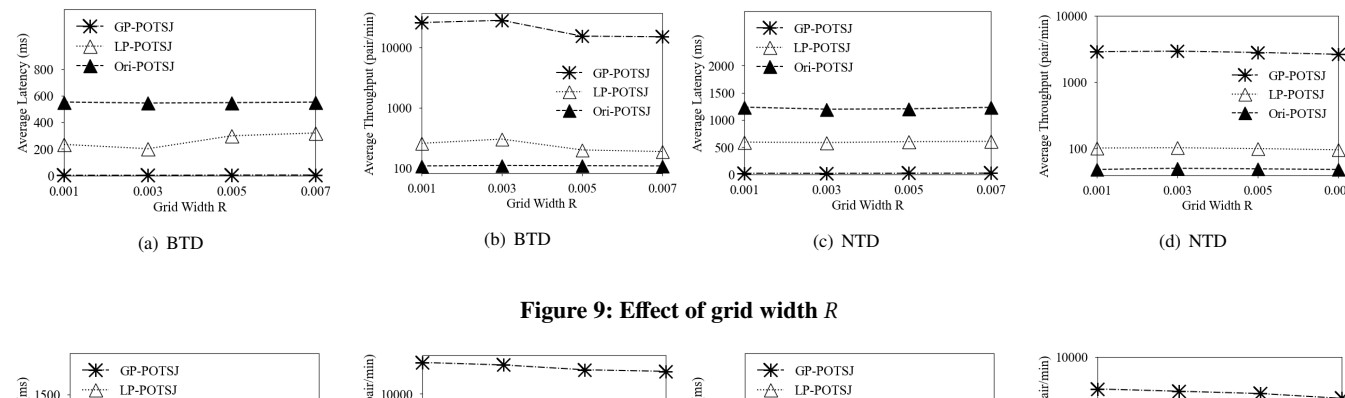

**Figure 9: Effect of grid width $R$**

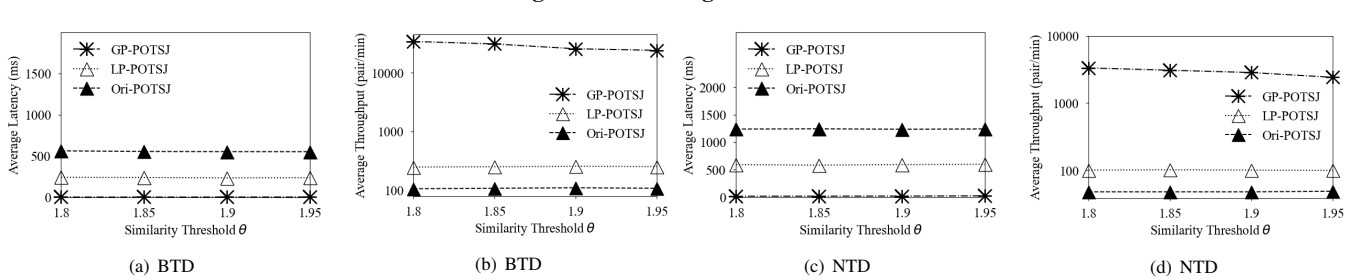

**Figure 10: Effect of similarity threshold $\theta$**

NTD containing mostly short trajectories. The grid-based pruning method primarily benefits longer trajectories. Selecting an optimal grid width between 0.001 and 0.005 enhances efficiency, as smaller widths generate too many grids, increasing pruning overhead, while larger widths raise the cost of searching within grids.

*Effect of similarity threshold $\theta$.* . We vary the similarity join threshold $\theta$ to assess its impact on POTSJ. As shown in Figure 10, GP-POTSJ, which uses global pruning, is sensitive to $\theta$. Its average latency increases slightly with higher $\theta$, while throughput decreases slightly. In contrast, LP-POTSJ and Ori-POTSJ, which lack global pruning, are unaffected by $\theta$, showing stable performance across metrics. This occurs because a lower $\theta$ allows early pruning of trajectory pairs, reducing the need for precise ST-similarity calculations.

*Effect of spatial weight $\alpha$.* . We vary the spatial weight to evaluate its effect on the framework. As shown in Figure 11, both average latency and throughput remain stable across different similarity weights. Additionally, pruning-based methods show a clear performance advantage. Overall, the framework is not sensitive to the parameter $\alpha$.

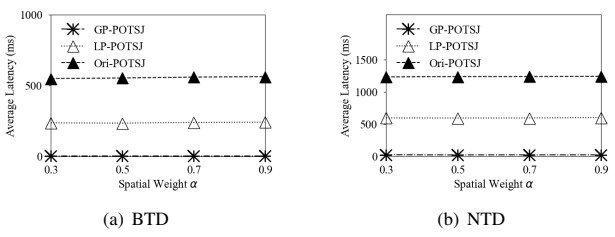

**Figure 11: Effect of spatial weight $\alpha$**

*Effect of exponential decaying parameter $\lambda$.* . We vary the exponential decay parameter $\lambda$ from Equation 4 to assess its impact on the framework. Figure 12 shows that average latency across all baselines remains stable as $\lambda$ changes, indicating the framework's insensitivity to this parameter. Thus, adjusting $\lambda$ can control the focus on recent data, allowing for flexibility in different processing scenarios.

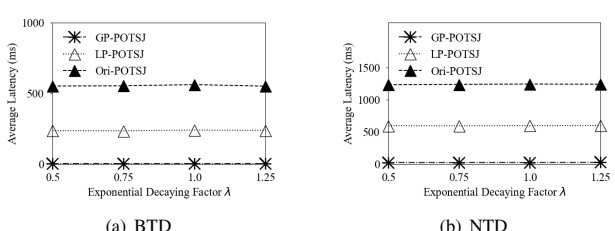

**Figure 12: Effect of exponential decaying parameter $\lambda$**

## C   Pseudo Code of Detailed Algorithms

### C.1   Dynamic Workload Adjustment

***Algorithm details.*** Algorithm 1 presents the pseudo code of the dynamic workload adjustment algorithm. The input consists of an incoming sample location $p_{\tau_i}$, trajectory to partition map $M$ containing the mapping relationship of all trajectories in $S_P$ and their corresponding partition columns, the workload array $L_{col}$ corresponding to each column, the column index used by previous cyclic partitioning $c_i$ and a threshold $\epsilon$. The output is an updated partition index $I^*$.

For simplicity, we consider a online self-join (i.e., $S_P = S_Q$) here. Note that our solution supports non-self joins as well. Firstly, if $M$ contains the partition column index corresponding to $\tau_i$, it indicates

**Algorithm 1:** Dynamic Workload Adjustment

**Data:** Sample location $p_{\tau_i}$, trajectory to partition map $M$, column workload array $L_{col}$, the column index used by previous cyclic partitioning $c_i$, unbalanced factor threshold $\epsilon$

**Result:** reallocated location partition index $I^*$

1 **if** $M.contains(\tau_i)$ **then**
2    $I^* \leftarrow M.get(\tau_i)$;
3 **else if** $UBF_{col} < \epsilon$ **then**
4    $I^* \leftarrow (c_i + 1)\%L_{col}.size()$;
5    $c_i \leftarrow c_i + 1$;
6    $M.put(\tau_i, I^*)$;
7 **else**
8    $I^* \leftarrow I_{min}$;
9    $M.put(\tau_i, I^*)$;
10 **for** *each partition load* $l^{(i)}$ *in* $L_{col}$ **do**
11    **if** $i = I^*$ **then**
12      $l^{(i)} \leftarrow l^{(i)} + L_{col}.size() + 1$;
13    **else**
14      $l^{(i)} \leftarrow l^{(i)} + 1$;
15 $UBF_{col} \leftarrow max(L_{col})/min(L_{col})$;
16 $I_{min} \leftarrow \arg\min_I L_{col}[I]$

that $\tau_i$ is not a new trajectory and $p_{\tau_i}$ should be assigned to the column corresponding to $\tau_i$ (Lines 1–2). Otherwise, it indicates that $\tau_i$ is a brand new trajectory. If $UBF_{col} < \epsilon$, the algorithm is in Phase 1 monitoring work load, and $p_{\tau_i}$ should be assigned to the column in a cyclic manner (Lines 3–6). Else, the dynamic load adjustment phase has been triggered and the algorithm enters Phase 2, $p_{\tau_i}$ should be assigned to the column with the lowest load $I_{min}$ (Lines 7–9). Subsequently, the workload of all column partitions is updated. In the case of self-join, a same sample location is distributed once in both column and row. The load of column to which $I^*$ belongs should be incremented by the matrix dimension plus 1, while work load of the remaining columns should be incremented by 1 (Lines 10–14). Finally, value of $UBF_{col}$ is updated, along with the index of the column with the smallest load $I_{min}$ (Lines 15–16).

## C.2 Overall Incremental Similarity Update

*Algorithm details.* Algorithm 2 presents the pseudo code of the overall incremental similarity update. This algorithm updates all Location-to-Trajectory similarities resulting from the generation of $p_{\tau_i}$. First, it updates the spatial and temporal similarities between $p_{\tau_i}$ and each trajectory $\tau_j$ (excluding $\tau_i$) using spatial neighborhood search (Line 2) and binary search (Line 5). Then, it updates the global upper bound of the spatio-temporal similarity between $\tau_i$ and $\tau_j$. If this upper bound is below the similarity threshold $\theta$, the pair $\tau_i$ and $\tau_j$ is unlikely to be similar and is added to the pruned set $A_p$ (Lines 6–8). Finally, it updates the similarity between $\tau_i$ and all other locations in the partition using dimensional linear pruning (Line 9).

## C.3 Spatial Neighborhood Search

*Algorithm details.* Algorithm 3 presents the pseudo code of the grid-neighborhood round search algorithm. Starting from the grid

**Algorithm 2:** Incremental Similarity Update

**Data:** Newly coming trajectory location $p_{\tau_i}$, the grid $p_{\tau_i}$ belongs to $G_{p_{\tau_i}}$, in partition sample location set $P$, in partition trajectory set $T$, in partition grid set $G$, round search depth $C$, grid width $R$, spatio-temporal similarity threshold $\theta$

**Result:** Pruned trajectory pair set $A_p$

1 $A_p \leftarrow \emptyset$;
2 $spatialNeighborhoodSearch(p_{\tau_i}, G_{p_{\tau_i}}, T, C, R)$;
3 **for** *each* $\tau_j \in T$ **do**
4    **if** $\tau_i \neq \tau_j$ **then**
5      $binarySearch(p_{\tau_i}, \tau_j)$;
6      compute $Sim_{st}(\tau_i, \tau_j).UB$;
7      **if** $Sim_{st}(\tau_i, \tau_j).UB < \theta$ **then**
8        $A.add(<\tau_i, \tau_j, false>)$;
9 $dimensionalLinearPrune(p_{\tau_i}, G, P)$;
10 **return** $A_p$;

**Algorithm 3:** Spatial Neighborhood Search

**Data:** Newly coming sample location $p_{\tau_i}$, the grid $p_{\tau_i}$ belongs to $G_{p_{\tau_i}}$, in partition trajectory set $T$, round search depth $C$, grid width $R$

**Result:** Update similarity set $Q$

1 $Q \leftarrow \emptyset; k \leftarrow 0; Set_\tau \leftarrow T; M \leftarrow \emptyset$;
2 **while** $k < C$ **do**
3    $G \leftarrow getGridsInDepth(k, G_{p_{\tau_i}})$;
4    **for** *each sample location* $p_{\tau_j}$ *in* $G$ **do**
5      **if** $\tau_j \in Set_\tau$ **then**
6        $d \leftarrow d_s(p_{\tau_j}, p_{\tau_i})$;
7        **if** $M.contains(\tau_j)$ **then**
8          **if** $d < M.get(\tau_j).d_s$ **then**
9            $M.remove(\tau_j)$;
10            $M.put(\tau_j, <p_{\tau_j}, d_s, false>)$;
11        **else**
12          $M.put(\tau_j, <p_{\tau_j}, d_s, false>)$;
13    **for** *each* $\tau_j$ *in* $M.keySet()$ **do**
14      **if** $M.get(\tau_j).d_s < min\left\{d_s(G_{p_{\tau_i}}, p_{\tau_i}) + k \times R\right\}$ **then**
15        $M.get(\tau_j).flag \leftarrow true$;
16        $Set_t.remove(\tau_j)$;
17    $k \leftarrow k + 1$;
18 **for** *each* $\tau_j$ *in* $M.keySet()$ **do**
19    **if** $M.get(\tau_j).flag = true$ **then**
20      $Q.add(<p_{\tau_i}, p_{\tau_j}, M.get(\tau_j).d_s>)$;
21 **for** *each* $\tau_j$ *in* $Set_\tau$ **do**
22    **if** $|\tau_j| \leq 40$ **then**
23      $Q.add(bruteForce(p_{\tau_i}, \tau_j))$;
24    **else**
25      $Set_j = getCandidateGrids(\tau_j)$;
26      $Q.add(Search(p_{\tau_i}, Set_j))$;
27 **return** $Q$;

where $p_{\tau_i}$ is located, initiate a round search. First, determine the set of grids $G$ to be searched in the $k$-th round (Line 3). Then, traverse all sample locations in $G$ to find the closest location to $p_{\tau_i}$ in each trajectory, record the shortest distance, and mark it as false (Lines 4–12). Check if the location $p_{\tau_j}$ is the closest in $\tau_j$ to $p_{\tau_i}$. If the distance from $p_{\tau_j}$ to $p_{\tau_i}$ is less than the radius of the smallest circle centered at $p_{\tau_i}$ tangent to a grid edge in $G$, mark $p_{\tau_j}$ as the closest location in $\tau_j$ (Lines 13–16). Repeat this process until $k = C$ (Line 17). For unprobed trajectories, handle short trajectories by directly using brute force to find the closest location in $\tau_j$ and add it to set $Q$ (Lines 22–23). For long trajectories, draw a minimum circular region centered at $p_{\tau_i}$ that fully covers at least one grid passed through by $\tau_j$. Filter out all grids within this region and traversed by $\tau_j$, then search for the closest location in $\tau_j$ to $p_{\tau_i}$ in this candidate grid set and add it to set $Q$ (Lines 24–26).

## C.4 Dimensional Linear Prune

*Algorithm details.* Algorithm 4 presents the pseudo code of the dimensional linear pruning algorithm. First, we compute the perpendicular bisectors between $p_{\tau_i}$ and all other sample locations in $\tau_i$ and store them in set $H$ (Lines 2–3). Then, we calculate the lower bound ($LB$) and upper bound ($UB$) of the timestamp for the location $p_{\tau_j}$ to be updated (Lines 4–7). Next, we iterate through all grids (Line 8). If a grid lies on the same side of all bisectors in $H$ as $p_{\tau_i}$ (Line 9), all sample locations in that grid require updating. If a grid is on the opposite side, no update is needed. If a grid intersects any bisector in $H$, we verify the spatial relationship between its sample locations and $p_{\tau_i}$. Locations on the same side as $p_{\tau_i}$ are updated and added to the result set (Lines 10–12). Finally, we update the temporal Location-to-Trajectory distance from locations with timestamps between $LB$ and $UB$ with respect to $p_{\tau_i}$ (Lines 13–15).

---

**Algorithm 4:** Dimensional Linear Prune

**Data:** Newly coming trajectory location $p_{\tau_i}$, in partition grid set $G$, in partition sample location set $P$
**Result:** Update similarity set $Q$

1   $Q \leftarrow \emptyset$; $H \leftarrow \emptyset$; $LB \leftarrow -1$; $UB \leftarrow +\infty$;
2   **for** *each $p_{(\tau_i,k)} \in \tau_i$* **do**
3     $H$.add(mid perpendicular of $p_{\tau_i}$ and $p_{(\tau_i,k)}$);
4     **if** $p_{(\tau_i,k)}.t \le p_{\tau_i}.t$ **then**
5       $LB = max(LB, \frac{p_{\tau_i}.t - p_{(\tau_i,k)}.t}{2})$;
6     **else**
7       $UB = min(UB, \frac{p_{(\tau_i,k)}.t - p_{\tau_i}.t}{2})$;
8   **for** *each $g \in G$* **do**
9     **if** *$g$ and $p_{\tau_i}$ is on the same side of $H$ or $g$ is overlapped by $H$* **then**
10       **for** *each sample location $p_{\tau_j}$ in $g$* **do**
11         **if** *$p_{\tau_j}$ and $p_{\tau_i}$ is on the same side of $H$* **then**
12           $Q$.add($< p_{\tau_j}, p_{\tau_i}, d_s(p_{\tau_j}, p_{\tau_i}) >$);
13   **for** *each $p_{\tau_j} \in P$* **do**
14     **if** *$p_{\tau_j}.t \ge LB$ and $p_{\tau_j}.t \le UB$* **then**
15       $Q$.add($< p_{\tau_j}, p_{\tau_i}, d_t(p_{\tau_j}, p_{\tau_i}) >$);
16   **return** $Q$;

---

## C.5 Similarity Merge By Interval Update

We present the pseudo-code of approximate algorithm in Algorithm 5. Each new data entry triggers similarity updates and system clock progression. We establish the first time interval upon partition initialization, and subsequently, a new interval is created at preset intervals (Lines 2–5). Thus, each Location-to-Trajectory similarity pair belongs to a unique interval. When similarity update occurs for a Location-to-Trajectory pair cached before, the outdated similarity is removed from the original interval, and the updated similarity is added to the corresponding new interval (Lines 6–14). Additionally, after the clock progresses, we need to determine if any interval expiration. If expiration occurs, all data from the old interval is cleared (Lines 17–18). After the interval operations, we calculate the decaying factor corresponding to each interval, as well as the sum of all Location-to-trajectory similarities within each interval (Lines 15–20). Then spatio-temporal similarity can be obtained using Equation 10.

---

**Algorithm 5:** Similarity Merge By Interval Update

**Data:** New Location-to-Trajectory similarity $s(p_{\tau_i}, \tau_j)$, interval array $A_I$, interval length $l$, interval expiration threshold $\sigma$
**Result:** Updated Similarity $S$

1   $S \leftarrow 0$; $c \leftarrow$ current clock;
2   **if** $A_I.size() = 0$ **then**
3     $A_I.add$(new Interval from $c$ to $c + l$);
4     $A_I.tail().add(s(p_{\tau_i}, \tau_j))$;
5     **return** $s(p_{\tau_i}, \tau_j) * getDecayFactor(\frac{l}{2})$;
6   **else**
7     **if** $A_I.isExists(s(p_{\tau_i}, \tau_j))$ **then**
8       $A_I.getInterval(s(p_{\tau_i}, \tau_j)).remove(s(p_{\tau_i}, \tau_j))$;
9     **if** $c < A_I.tail().startTime$ **then**
10       $A_I.tail().add(s(p_{\tau_i}, \tau_j))$;
11     **else**
12       $c_e \leftarrow A_I.tail().endTime$;
13       $A_I.add$(new Interval from $c_e$ to $c_e + l$);
14       $A_I.tail().add(s(p_{\tau_i}, \tau_j))$;
15     **for** *each Interval $i$ in $A_I$* **do**
16       $df \leftarrow getDecayFactor(c - \frac{i.endTime - i.startTime}{2})$;
17       **if** $df < \sigma$ **then**
18         $A_I.remove(i)$;
19       **else**
20         $s \leftarrow s + df \times Interval.sum()$;
21   **return** $S$;

---

