# OpenReview forum: "Parallel Online Similarity Join over Trajectory Streams"
_ACM.org/TheWebConf/2025/Conference — WWW 2025 Poster_

### Official Review · Reviewer_T2Mr · 2024-11-10

**Novelty:** 2
**Technical Quality:** 2

**Review:**

Pros:

P1. The motivation of the paper (i.e., the online trajectory similarity join) is convincing and practical.

P2. The authors have crafted an easy-to-read introduction section, with a particularly vivid example that effectively illustrates the background and motivation of their research work.

P3. The authors have utilized two real-world datasets to evaluate their methods.

Cons:

C1. Compared to existing methods, I think the innovativeness of the authors' approach is somewhat lacking.

C2. The writing in the technical section requires improvement.

C3. The experimental section is not very convincing.


For specific comments, please refer to the subsequent Questions section.

**Questions:**

Q1. The authors mention in the introduction of existing works that they need to use a time window, and that the length of this time window is difficult to set. However, their novel approach employs an exponential-decaying method, which also requires setting a decay parameter value. This parameter value can also be interpreted as a form of time window length, as trajectories from earlier times have negligible impact on similarity due to decay. Please discuss this further and distinguish the differences between the proposed work and existing works.

Q2. Please explain the rationale behind the newly introduced exponential-decaying setting. Specifically, why was the exponential-decaying setting chosen? Also, discuss whether other settings could achieve a similar decaying effect and why the exponential-decaying setting was selected.

Q3. The parameters in the paper are somewhat confusing. For example, in Definition 1 and Definition 5, the subscript for p has only one value, but in Definition 2, p has two subscripts.

Q4. The similarity calculation in Definition 6 seems to be based solely on each point in the trajectory, without considering the order relationship between points within the trajectory. This appears to be unreasonable. Suppose there are three trajectories: t1 = {p1, p2, p3}, t2 = {p1, p2, p4}, and t3 = {p1, p4, p2}, where p1, p2, p3, and p4 are four distinct points. According to the similarity calculation method in Definition 6, the similarity between t1 and t2 is the same as that between t1 and t3, which is not reasonable.

Q5. At the end of Section 2.2, the authors mention that their method can be extended to other popular similarity measurements. Please elaborate on this.

Q6. The authors mention several lemmas in the paper. Please provide the semantic meaning of these lemmas before and after their lemmas.

Q7. The authors mention some pruning methods and associated lemmas. However, these lemmas only demonstrate the safety of pruning. Can the authors prove the effectiveness of pruning, specifically its impact on improving time complexity?

Q8. Please explain the reasons for the parameter settings in Table 1, i.e., why these values were chosen.

Q9. The authors introduce many existing related works, but only compare against [11] in the experiments. Please include comparison experiments with other existing works.

Q10. In Table 1, the authors state that the value of $\theta$ in the NTD dataset experiments ranges from 1.00 to 1.95. However, in Figures 10(c) and (d) in Appendix B, the value of $\theta$ is 1.8 to 1.95. Please explain the reason for this inconsistency.

**Reviewer Confidence:**

3: The reviewer is confident but not certain that the evaluation is correct

**Scope:**

3: The work is somewhat relevant to the Web and to the track, and is of narrow interest to a sub-community

---

### Official Review · Reviewer_nGUk · 2024-11-29

**Novelty:** 3
**Technical Quality:** 3

**Review:**

The paper proposes a framework to process online similarity based joins between two streams of trajectories. Motivation of the research mainly lies in that previously similar trajectories without recent updates may become dissimilar to each other and thus they should be identified and eliminated from the current result. Based on this motivation, the paper proposes a time-decaying similiarity function and design multi-threaded framework to parallelize similarity join of trajectories in multiple partitions. Load balancing and incremental similarity update are also discussed. Experiments are conducted on two real datasets. The results show the full approach from this paper outperforms its variants and one existing approach.

Strong points:
1. The paper gives some visual illustrations some of which are good.
2. The experiments use real datasets.
3. The idea of parallelizing trajectory join is good.

Weak points:
1. The motivation is problematic.
2. The writing could be improved.
3. The experiments are not convincing and should include more competitor approaches.

Detailed comments:
D1. The motivation is only half right. The situation could be fall in the other half: the two trajectories are still similar though they don't update recently.

D2. Furthermore, previously dissimiliar pairs may become similar recently. Is this case considered and included? Which part of the design actually addresses this issue?

D3. In relation to D1 and D2, the use of time-decay in the similarity function may fail to capture the reality.

D4. The partition approach fails to consider data skewness that is often seen for spatial data like trajectories. This issue can substantially affect partitioning effect and load balancing.

D5. In general, it is not convincing to exclude sliding window based approaches. They should also be compared with in the experiments.

Minor comments:
M1. The example with Figure 1 is hard to follow.
M2. A better example should be used in Section 4.1. For instance, a few trajectories can be shown on a map, followed by the partition scheme in Figure 3.

**Questions:**

If the authors do want to respond, please refer to D1 to D5 above.

**Reviewer Confidence:**

4: The reviewer is certain that the evaluation is correct and very familiar with the relevant literature

**Scope:**

2: The connection to the Web is incidental, e.g., use of Web data or API

---

### Official Review · Reviewer_BJqE · 2024-12-01

**Novelty:** 6
**Technical Quality:** 5

**Review:**

This paper proposes a POTSJ framework to process online TS-Join problem while simultaneously ensuring real-time updates, comprehensive evaluation, and scalability. In the POTSJ framework, the matrix-based partition strategy and Two-phase dynamic Load strategy are used to partition the stream of trajectories and balance the workload of partitions. The global and local pruning strategy are utilized to incrementally update similarity, which significantly reduces the time complexity of similarity computation.

Strength:
* S1: This paper is well-arranged, and the writing is logical and easy to understand.
* S2: This paper proposes a novel Online Trajectory Similarity-Join (Online TS-Join) problem.
* S3: Experimental results verify the effectiveness, efficiency and parameter robustness of the proposed POTSJ.

Weakness:
* W1: Authors merely exemplify the drawbacks of existing in-memory online search algorithms in handling Online TS-Join problem.  The motivation of studying Online TS-join problem is unclear. Furthermore, it's relationship to the Web Conference should be further clearified.
* W2: The accuracy of the approximation algorithm is not compared with the exact similarity merging algorithm.
* W3: There does not appear to be a comparison of the accuracy achieved by different methods.

**Questions:**

* Q1: Is there a reliable theory for Interval Update (section 6.2) that the approximation algorithm through interval update does not lose too much accuracy?
* Q2. Why the authors did not compare the accuracy of approximation algorithm with exact similarity merging algorithm?
* Q3. Why the authors did  not compare the accuracy of POTSJ with the baseline methods and the ablation versions?

**Reviewer Confidence:**

3: The reviewer is confident but not certain that the evaluation is correct

**Scope:**

3: The work is somewhat relevant to the Web and to the track, and is of narrow interest to a sub-community

---

### Official Review · Reviewer_njmW · 2024-12-02

**Novelty:** 6
**Technical Quality:** 6

**Review:**

Interesting paper. Related to the conference/track. Timely topic. Important problem variation (real-time). Robust modelling and methodology. Quite holistic performance evaluation. Nice read!

**Questions:**

What about other methodologies for comparison, except for the "ghost" one?
What about other practical mobility-related performance metrics?

**Reviewer Confidence:**

4: The reviewer is certain that the evaluation is correct and very familiar with the relevant literature

**Scope:**

4: The work is relevant to the Web and to the track, and is of broad interest to the community

---

### Official Review · Reviewer_eVtJ · 2024-12-05

**Novelty:** 5
**Technical Quality:** 4

**Review:**

Strengths:
1. This study developed a matrix-based approach (coupled with multi-level pruning) to improve the effectiveness and scalability for online (real-time) parallel trajectory join problems over existing sliding window based approaches which are not scalable enough and cannot deal with time-aware decaying of outdated trajectory paths.
2. The paper is well-written, and the experimental design is quite comprehensive.
3. Use of multi-level pruning across both spatial and temporal dimensions which improves the efficiency of similarity calculations.

Weakness:
1. The use of matrix-based partitioning, while innovative, can face computational complexity issues. Specifically, the size of partitions requires parameter tuning, which could become a bottleneck as the dataset size increases or as parameters are scaled for diverse applications. This limitation could undermine the framework’s scalability and efficiency, particularly in dynamic or large-scale environments
2. The paper’s evaluation compares the proposed framework primarily with the GHOST [11] framework. However, the other three comparisons provided fall under the category of ablation analysis. This lack of rigorous comparison with multiple state-of-the-art frameworks weakens the validation of the framework’s superiority and general applicability. Including comparisons with a wider array of contemporary methods would make the results more robust and convincing.
3. The concept of time-aware exponential decay introduced in the framework demands more extensive validation. While theoretically sound, the paper does not sufficiently demonstrate how frequent or significant this issue is in real-world scenarios. Without concrete evidence from real-world datasets, the necessity and impact of this feature remain speculative.

**Questions:**

Issues:
1. The matrix size, like the window size, is also a static parameter that could harm scalability.
2. The three iterative versions of the proposed framework should be included in a separate ablation analysis rather than in the main experimental results.
3. Other than the Ghost framework, why do the authors refrain from making strong comparisons with other state-of-the-art methods?
4. Why does the experimental analysis include only these two datasets, as there remain some other datasets in the domain too?

**Reviewer Confidence:**

2: The reviewer is willing to defend the evaluation, but it is likely that the reviewer did not understand parts of the paper

**Scope:**

4: The work is relevant to the Web and to the track, and is of broad interest to the community